# Five-Year Incidence, Management, and Visual Outcomes of Diffuse Lamellar Keratitis after Femtosecond-Assisted LASIK

**DOI:** 10.3390/jcm10143067

**Published:** 2021-07-11

**Authors:** Majid Moshirfar, Kathryn M Durnford, Adam L Lewis, Chase M Miller, David G West, R Alek Sperry, William B West, Kathryn M Shmunes, Shannon E McCabe, MacGregor N Hall, Yasmyne C Ronquillo, Phillip C Hoopes

**Affiliations:** 1Hoopes Vision, HDR Research Center, 11820 S. State Street Suite #200, Draper, UT 84020, USA; kms04@gmail.com (K.M.S.); smccabe@hoopesvision.com (S.E.M.); yronquillo@hoopesvision.com (Y.C.R.); pch@hoopesvision.com (P.C.H.); 2John A. Moran Eye Center, Department of Ophthalmology and Visual Sciences, Salt Lake City, UT 84132, USA; 3Utah Lions Eye Bank, Murray, UT 84107, USA; 4School of Medicine, University of Utah, Salt Lake City, UT 84132, USA; kathryn.durnford@hsc.utah.edu (K.M.D.); William.West@hsc.utah.edu (W.B.W.J.); 5College of Osteopathic Medicine, Kansas City University, Kansas City, MO 64106, USA; adamlewis@kansascity.edu; 6McGovern Medical School, University of Texas Health Science Center at Houston, Houston, TX 77030, USA; chase.m.miller@uth.tmc.edu (C.M.M.); macgregor.hall@uth.tmc.edu (M.N.H.); 7Brigham Young University, Provo, UT 84602, USA; gillison.west1@gmail.com (D.G.W.); rasperry@byu.edu (R.A.S.); 8Department of Ophthalmology, College of Medicine-Jacksonville, University of Florida, Jacksonville, FL 32209, USA; 9Mission Hills Eye Center, Pleasant Hill, CA 94523, USA

**Keywords:** diffuse lamellar keratitis (DLK), femtosecond, LASIK, incidence

## Abstract

Femtosecond (FS) lasers initially had a higher incidence of diffuse lamellar keratitis (DLK) compared with microkeratome flap creation. It has been theorized that higher-frequency lower-energy (HFLE) FS lasers would reduce the incidence of DLK. Our study sought to evaluate the incidence of newer HFLE FS lasers with pulse frequencies above 60 kHz. It was a retrospective case-control study evaluating the incidence of DLK following flap creation with one of three FS lasers (AMO iFs, WaveLight FS200, Zeiss VisuMax). Uncomplicated LASIK cases were included as the control group (14,348 eyes) and cases of DLK were recorded in the study group (637 eyes). Of the 637 cases of DLK, 76 developed stage II, 25 progressed to stage III, and only three developed stage IV DLK. The overall incidence rate of DLK was 4.3%; it has fallen with the invention of newer HFLE FS lasers and is approaching the DLK incidence rates of DLK with microkeratome.

## 1. Introduction

Laser in situ keratomileusis (LASIK) is a popular and safe option for the correction of refractive errors in the United States and worldwide. It is the most common refractive surgery performed, and overall complication rates since 2010 are less than 0.8% [1,2]. Patient satisfaction has been shown in multiple studies to exceed 90% [3]. LASIK procedures typically involve creating a corneal flap, which may lead to intraoperative complications, such as flap tears, incomplete flaps, epithelial defects, and interface debris [4]. Postoperative complications include flap dislocation, flap striae and folds, diffuse lamellar keratitis (DLK), and central toxic keratopathy (CTK) [4]. DLK is a common complication of LASIK, and recent studies report incidences after femtosecond laser-assisted LASIK ranging from 0.5 to 37.5% [5,6,7,8,9,10,11,12,13,14]. DLK has been described as a serious and sight-threatening condition that requires prompt treatment with close follow-up to avoid permanent consequences of stromal tissue loss, corneal scarring, astigmatism, hyperopic shift, and decreased visual acuity [15].

Corneal flaps created with femtosecond (FS) lasers have been observed to predominantly induce keratocyte necrosis at higher rates than microkeratome blade, which has been observed to induce higher levels of apoptosis [6,9,16]. Necrosis leads to an uncontrolled release of intracellular components, attracting more leukocytes and generating a more intense inflammatory response [14]. The more intense inflammatory response is believed to explain the higher rates of DLK, an inflammatory response, seen in FS lasers as compared with the microkeratome blade for flap creation [9].

IntraLase (Irvine, California), with pulse frequencies of 10 kHz, was the first FS laser to market following FDA approval of their first-generation laser in 2001. Since that time, FS laser technology has improved, and newer FS lasers are capable of high-frequency and low-energy (HFLE) compared with the earlier generation of low-frequency high-energy (LFHE) lasers [17]. The FS laser technology allows for surgeons to have more precision and more predictability in flap creation as compared with the mechanical microkeratome blade [9]. The frequencies that these HFLE FS lasers have been able to reach are theorized to reduce the amount of corneal inflammation and thereby reduce the incidence of DLK [18].

Netto et al. found that in an animal model, higher pulse frequencies at lower energy resulted in roughly a 50% reduction in corneal monocyte infiltration and almost a 50% reduction in corneal cell death at 60 kHz compared with the higher energy 15 kHz laser [18]. However, Choe et al. found no significant difference in the incidence of DLK in patients at 9.7, 13.3, and 14.0% between the 15, 30, and 60 kHz FS lasers, respectively [8]. Another study reported DLK incidence as low as 0.5%, lower than the incidence some have reported for DLK in microkeratome LASIK [7]. Unfortunately, there has been a paucity of data in recent years related to the incidence of DLK with newer, even lower energy FS lasers. The purpose of this study is to report the incidence of DLK in patients who underwent femtosecond LASIK over five years at a single site using newer HFLE FS lasers at 150, 200, and 500 kHz frequencies. This study also sought to describe visual outcomes and management of DLK in these newer FS lasers.

## 2. Materials and Methods

### 2.1. Study Design

This retrospective study reviewed the records of consecutive myopic, hyperopic, and astigmatic LASIK procedures performed on 14,985 eyes using the Zeiss VisuMax (Carl Zeiss Meditec, Oberkochen, Germany), AMO iFS (Abbott Medical Optics, Santa Ana, CA, USA), or WaveLight FS200 (Alcon Laboratories, Fort Worth, TX, USA) FS laser at a single site from January 2015 to December 2019. Stromal ablation was performed by the WaveLight EX500 (Alcon Laboratories, Fort Worth, TX, USA) laser in all cases. After initial data gathering identified 22,053 surgeries from a central database, all duplicates were removed and any patients who developed CTK following LASIK (*n* = 12) were removed as well. Enhancement procedures occurring after the initial LASIK were not included in either the cases or control groups so that all reports were of only initial refractive surgery. Additionally, any LASIK performed combined with an additional procedure such as an implantable collamer lens was removed from the study population. This left a total of 14,985 eyes of 7867 patients. Of these patients, 7118 received binocular LASIK and 749 underwent monocular LASIK. Of all the LASIK patients, 637 eyes were diagnosed with DLK during the postoperative period and the remaining 14,348 surgeries made up the control cohort. This study was approved by the Hoopes Vision Ethics Board and adheres to the tenets outlined in the Declaration of Helsinki. The study was HIPAA-compliant with informed consent obtained from all patients. It was an IRB-approved study by the Biomedical Research Alliance of New York (BRANY, Lake Success, NY, USA) in accordance with research standards and state law.

### 2.2. Surgical Technique

The corneal flap was created with Zeiss VisuMax, AMO iFs, or WaveLight FS200 FS lasers. The AMO iFs, WaveLight FS200, and VisuMax had pulse frequencies of 150, 200, and 500 kHz, respectively. The flap thickness was set to 100 or 110 μm for all three FS lasers. Raster energy was 0.85 or 0.95 μJ for the AMO iFs laser, 0.8 μJ for the WaveLight FS200, and 0.6 μJ for the VisuMax laser. The side cut energies were 0.6 μJ for the AMO iFs and WaveLight FS200 lasers and 0.8 μJ for VisuMax laser. Superior hinges were created in all eyes. The right eye was always treated before the left eye in bilateral cases.

### 2.3. Postoperative Care

The standard postoperative treatment regimen for all patients following LASIK included prednisolone acetate 1% eye drops every 2 h for the 1st 24 h and then every 4 h for the remainder of the 1st postoperative week. Topical fourth-generation fluoroquinolones were prescribed four times per day for the 1st postoperative week. For patients in whom DLK was diagnosed, they were promptly treated and seen with frequent follow-up visits. The grade of DLK was determined clinically according to the staging outlined by Linebarger et al. [15]: stage I—white granular cells in the flap periphery; stage II—white granular cells at the flap center; stage III—white cells more clumped in the flap center, often with relative clearing in the periphery; stage IV—stromal melting, central striae, bullae formation, and scarring.

The treatment was surgeon-dependent and was tailored according to the stage of DLK. Increased frequency of the topical steroid drop was implemented in the vast majority of patients. Steroid drops were prescribed every 1–4 h for stage I and 1–2 h for higher grades. When patients exhibited stage III symptoms or there was a concern for progression to stage III, irrigation was considered to decrease the risk of progression to stage III/IV. Along with the concern of progression to subsequent stages of DLK, additional treatments, such as vitamin C and other supplements, were started for DLK stages II–IV when there was concern for the development of CTK. Oral doxycycline was added for later stages of DLK to treat systemic conditions, such as rosacea, that are thought to predispose patients to DLK.

### 2.4. Measured Variables

Preoperative clinical data collection for all patients included the date of LASIK procedure, age, sex, uncorrected distance visual acuity (UDVA), corrected distance visual acuity (CDVA), and manifest refraction. In addition, postoperative UDVA, CDVA, and manifest refraction were recorded at 1, 3, 6, and 12 months after LASIK. For patients who developed DLK, the day of onset and resolution were recorded in addition to the stage, UDVA, CDVA, manifest refraction, slit-lamp examination findings, and treatment initiated during each follow-up visit for DLK. The variables used to compare the cases and control groups included age, sex, and preoperative manifest refraction.

Statistical analysis was performed using the R statistical software, version 4.0.2 (22 June 2020). Comparisons of demographics and risk factors for the cohort that developed DLK and the cohort without DLK were performed using *t*-tests for continuous and Pearson’s chi-squared test or Fisher’s exact test when appropriate for categorical variables. Univariate logistic regression models were used to assess the relationship between risk factors and the time to resolution. A *p* value < 0.05 was considered statistically significant.

## 3. Results

The average age of the study cohort was 33.7 ± 8.25 (SD) years and ranged from 18 to 57 years. Table 1 shows preoperative and demographic differences between the patients that developed DLK and those who did not develop DLK. There were no statistically significant differences between the control group and the patients who developed DLK.

Following LASIK, DLK developed in 637 eyes of 436 patients, of which 235 cases were unilateral DLK and 201 were bilateral DLK. All cases developed within the 1st month following LASIK, with the majority diagnosed in the 1st week (range: 0–20). Unilateral DLK occurred after unilateral LASIK in 235 cases; the remaining 201 cases occurred after bilateral LASIK. The overall incidence of DLK was 4.3%.

The majority (533, 83.7%) of the DLK cases were stage I. As expected, the incidence declined with increasing severity: 76 eyes (11.9%) progressed to stage II DLK, while only 25 eyes (3.9%) advanced to stage III DLK, and only three eyes (0.5%) developed stage IV DLK. Out of the 906, 65 (7.2%) eyes that underwent LASIK using the WaveLight FS200 laser developed DLK, while only 528 of 12,512 eyes (4.2%) and 44 of 1567 eyes (2.8%) developed DLK in AMO iFs and VisuMax cases, respectively. Table 2 shows the numbers and percentages of eyes that developed various stages of DLK according to the FS laser used for flap creation. Figure 1 shows the proportion of DLK by stage for each FS laser. There was no significant difference between the laser types and development of different stages of DLK.

Table 3 shows the demographics comparison between stages of DLK. There was no significant difference in age, gender, affected eye, or laser type compared with the stage of DLK. The time to onset was somewhat comparable between the four groups and lacked any statistically significant difference between the four stages. The time to resolution was trending towards significance (*p* = 0.063) with a pattern showing the mean time to resolution increasing with stage severity.

Nearly every patient (98.9%) had an increased steroid dosing schedule following diagnosis. Only two patients did not receive additional treatments because their presentation of DLK stage I was minimal; both cases resolved within six days of onset. Higher potency steroids or oral steroids, oral antibiotics, supplements, gel tears, and bandage contact lenses were used significantly more frequently to treat stages II/III (*p* ≤ 0.001). In addition, irrigation was performed for more patients in stage III than other stages (*p* ≤ 0.001). Table 4 lists all treatments provided for DLK cases by stage.

Figure 2 evaluates the visual outcomes between the cases of DLK and their controls. The presence of DLK was associated with an increased likelihood of patients having UDVA of worse than 20/20 in the 1st postoperative month (OR = 1.33 (1.06, 1.667), *p* = 0.012). This difference resolved by the end of the 1st postoperative year (OR = 0.99, (0.73, 1.31), *p* = 0.922). The difference in visual acuity outcomes was most apparent in patients with high-grade DLK; the OR for UDVA worse than 20/20 was 2.4, 1.81, and 2.86 for stages II, III, and IV, respectively, stage II being significant (*p* < 0.001). Though trending towards significance, higher grades of DLK were not associated with worse visual outcomes at 12 months postoperatively. This lack of significance in the more severe stages III/IV can be most likely attributed to the smaller population of DLK patients in comparison with the number of stages I/II patients.

The standard graphs for reporting refractive surgery outcomes at twelve months are shown in Figure 3 to evaluate the visual outcomes of DLK patients. There was an increase in the rates of DLK occurring from October 2017 to January 2018 when assessing the trend of incidence of DLK over the five years.

## 4. Discussion

Following the introduction of FS lasers, it was noted that there was an increase in the rates of DLK compared with microkeratome [6,9]. Previous studies have evaluated DLK incidence across a range of pulse frequencies. The incidences of DLK for the LFHE FS lasers with pulse frequencies from 15to 60 kHz ranged from 0.5 to as high as 37.5%, with a mean of 14.2% ± 10.0 (SD) [5,6,7,8,9,10,11,12]. In the animal model, 60 kHz was a high enough frequency to produce significantly lower corneal inflammation and cell death rates than the 15 kHz FS laser [18]. Choe et al. did not observe this same impact when comparing 15, 30, and 60 kHz lasers among patients [7,8].

Since then, studies have evaluated the incidence of DLK in even lower energy FS lasers with pulse frequencies ranging from 150 to greater than 5000 kHz. In 2013, Tomita et al. was the first to report on the newer FS lasers and noted that the DLK rates of Zeimer’s Femto LDV system that could operate at 1000 kHz were around 8.2% [11]. Following that, a study was published on VisuMax with a frequency of 500 kHz with rates of 6.6% [13]. Most recently, a study noted DLK rates of 0.6% with Zeimer’s LDV Z2 and Z4 lasers that can operate above 5000 kHz. Table 5 reviews the current published literature on DLK incidences with various FS laser settings. For laser pulses of 60 kHz or less, the mean DLK rate is 14.2% ± 10 (SD), whereas the mean DLK rate for laser frequencies of 150 kHz and above is 4.9% ± 2.9 (SD). The mean raster energy of the LFHE lasers is 1.67 ± 0.58 (SD) compared with the mean raster energy of the HFLE lasers at 0.60 ± 0.36 (SD).

Our study reflected a lower incidence of DLK of the AMO iFs, WaveLight FS200, and VisuMax FS lasers at 4.2, 7.2, and 2.8%, respectively. The mean incidence of these newer HFLE FS lasers was 4.9% ± 2.9 (SD). Additionally, the mean raster energy between the LFHE and HFLE lasers dropped almost three times, from 1.67 to 0.60, matching the three-fold drop in the incidence rates. Earlier generations of FS lasers saw increased rates of DLK compared with the microkeratome, but studies of the later HFLE FS lasers have demonstrated a reduction in the incidence of DLK.

The reported incidences of DLK of the LFHE lasers were initially higher than microkeratome LASIK [6,9]. FS lasers allow for far more customization and control than microkeratomes and so this increase in DLK incidence was outweighed by these benefits. Now, the DLK rates of the HFLE lasers are approaching that of the microkeratome. A large study of 15,119 eyes showed a DLK incidence rate of 0.4% in microkeratome [19]. Of note, a study looking at the incidence of DLK in microkeratome procedures included one of the surgeons and authors of this current study. This study found an incidence rate for DLK of 6.2% in the microkeratome group (*n* = 896) [9]. The range of incidence of DLK between microkeratome and FS lasers overlaps; the newer HFLE FS lasers are as safe or safer than the microkeratomes regarding DLK incidence with the added benefit of more flap customization.

A majority of cases of DLK were stage I and II (95.6%) and resolved within ten days (Table 3). The time to resolution between stages was trending towards significance with a *p* value of 0.063 with a trend that the mean time to resolution increased with increasing severity. Treatments varied based on the severity of DLK at presentation and persistence of DLK (Table 4). Steroid treatments with a combination of antibiotics, supplements, and ocular surface protection were effective treatments. Oral doxycycline was also used to treat later stages more frequently than stage I DLK (*p* < 0.001). Vitamin C and coenzyme Q10 were also used in stages III/IV to prevent progression to CTK (*p* < 0.001). Ocular surface treatments using bandage contact lenses and gel tears were given more frequently to stages II/III as part of the DLK treatment (*p* < 0.001). The majority (97%) of patients’ DLK resolved within the 1st month, with most resolving within two weeks. Oral steroids and higher potency steroids were used significantly more frequently for stages II/III (*p* < 0.001). Irrigation was used at a significantly higher rate (*p* < 0.001) for stage II/III to prevent further progression of DLK.

The presence of DLK was associated with an increased likelihood of patients having UDVA of worse than 20/20 in the 1st postoperative month (OR = 1.33 (1.06, 1.667), *p* = 0.012). This difference seemingly resolved by the end of the 1st postoperative year (OR = 0.99, (0.73, 1.31, *p* = 0.922). The difference in visual acuity outcomes was most apparent in patients with high-grade DLK; the OR for UDVA < 20/20 was 2.4, 1.81, and 2.86 for stages II, III, and IV, respectively, with stage II being significant (*p* < 0.001). All patients with DLK achieved CDVA of 20/40 or better by their one-month postoperative visit. Despite having developed DLK, there was no lasting visual impact or difference in outcome between the LASIK patients that did not develop DLK. The visual outcomes in Figure 3 show that the vast majority (99%) of patients with DLK achieved a UDVA of 20/40 or better in the postoperative period. At 12 months, only 4% of patients lost one line of CDVA, and no patient lost more than two or more lines from the Snellen chart. Visual outcome stability can be seen in Figure 3F, where the mean spherical equivalent remains relatively the same from about one month postoperatively. For the majority of patients, the angle of error at 3 and 12 months lay between −5° and 5° with a limited number of patients outside of −15° and 15° of axis (Figure 3I). In addition, Figure 2 shows that despite having an increased likelihood of having UDVA worse than 20/20 in the 1st postoperative month, this difference resolved by the end of the 1st postoperative year with no statistically significant difference in visual outcomes between the different stages either. 

The etiology of DLK is not well-elucidated but is usually described as either sporadic or epidemic [20]. Sporadic cases of DLK are usually isolated and triggered by intraoperative causes such as epithelial defects [21] or retained blood in the interface [20,22], but can also be caused by patient factors, such as atopic diseases [23], and inflammatory conditions, such as ocular rosacea [24], meibomian gland dysfunction [25], and viral keratitis [26]. Epidemic cases of DLK occur in clusters and have been linked with exogenous factors, such as surgical marking pens [27], surgical gloves [28], and bacterial endotoxins in sterilization reservoirs or solutions [29,30]. Improper air purification has also been implicated [31]. The authors attributed the apparent epidemic flare in the present study to either the air filters, recent carpet cleaning, or changing the solvent used to clean the surgical tools, which was described at length in a recent study evaluating CTK occurrence within the same population [32]. After the clinical site adjusted the air filters and surgical cleaning solvent, cases of DLK subsequently fell.

Besides the rise in cases that we have attributed to the epidemic, we did not find preoperative risk factors or treatment differences associated with an increase in the risk of DLK. There was no significant impact of age, sex, or FS laser type between stages of DLK. Laser type or laser pulse frequencies of 150, 200, and 500 kHz did not significantly impact which patients developed DLK.

Our results may be limited by the study’s retrospective nature and the related limits of standardization, but this is the only study to evaluate the incidence of DLK for the AMO iFs and WaveLight FS200 lasers. We cannot exclude other risk factors, such as atopic disease, meibomian gland dysfunction, ocular rosacea, etc., that have been implicated in sporadic cases of DLK as factors that may have predisposed some individuals to the onset of DLK. We recommend that studies of the HFLE lasers continue to evaluate incidence rates of DLK of the various pulse frequencies to identify how closely correlated higher frequencies are to the incidence of DLK. As of yet, it has not been discussed if continuing to raise pulse frequency will further lower rates of DLK or if there is a maximum frequency needed before DLK incidence due to FS laser is minimized to the greatest extent possible. Ultimately, our results showed evidence of a decrease in the incidence of DLK that is consistent with studies done with other HFLE FS lasers.

## 5. Conclusions

In summary, the incidence of DLK appears to be falling as lower energy FS lasers have become available, which is consistent with the hypothesis that lower energy settings would reduce DLK incidence secondary to a reduction in cell death and corneal inflammation. Our study showed that newer generations of HFLE FS lasers are both safe and effective to use.

## Figures and Tables

**Figure 1 jcm-10-03067-f001:**
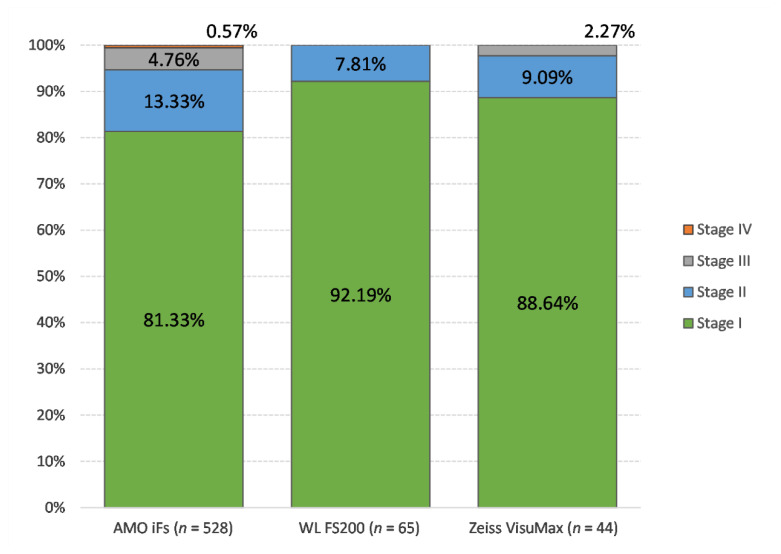
Comparison of DLK stages between laser types.

**Figure 2 jcm-10-03067-f002:**
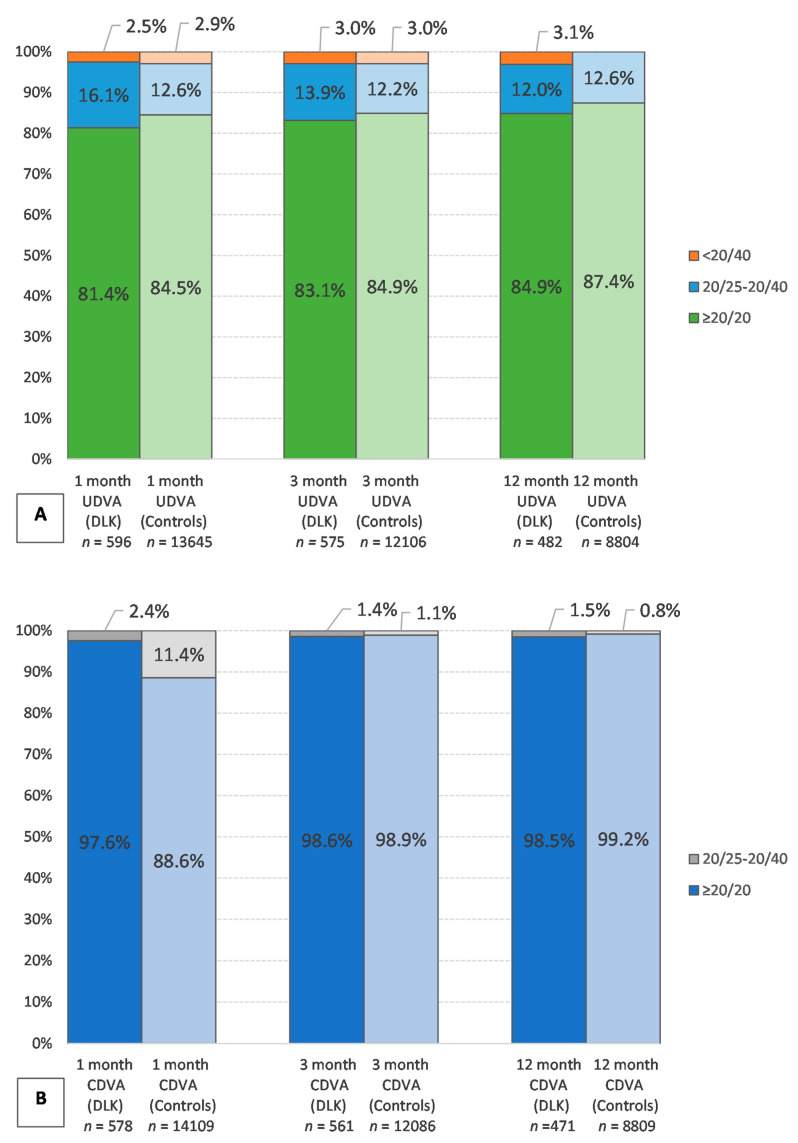
Visual outcomes of controls compared with DLK cases at 1, 3, and 12 months. (**A**) Comparison of UDVA over 1–12 months postoperatively. (**B**) Comparison of CDVA over 1–12 months postoperatively. DLK—diffuse lamellar keratitis; UDVA—uncorrected distance visual acuity; CDVA—corrected distance visual acuity.

**Figure 3 jcm-10-03067-f003:**
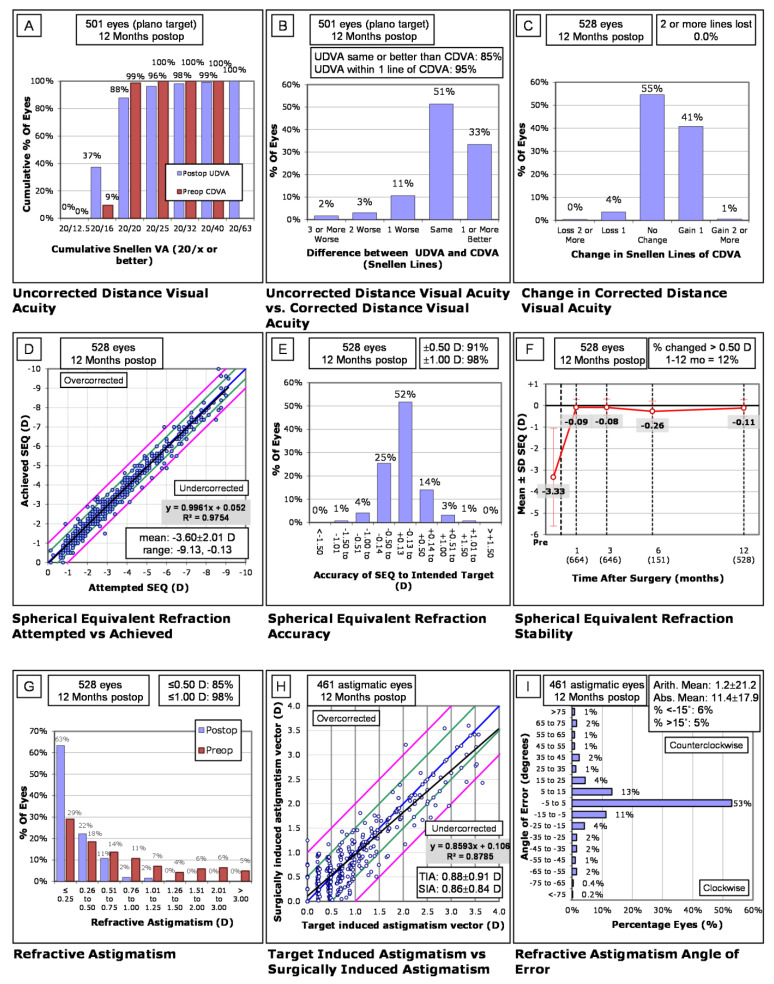
Analysis of DLK patients at 12 months postoperatively. (**A**–**I**) Standard graphs for refractive surgery shown above. (**A**) Uncorrected distance visual acuity (UDVA) at 12 months. (**B**) The difference in UDVA and corrected distance visual acuity (CDVA) at 12 months postoperatively. (**C**) Change in postoperative CDVA compared with preoperative CDVA. (**D**) Comparison of the attempted refractive spherical equivalent (SEQ) with the achieved refractive SEQ. (**E**) The accuracy of the attempted SEQ with the achieved SEQ. (**F**) Stability of the refractive SEQ postoperatively over 12 months. (**G**) Refractive astigmatism. (**H**) Target induced astigmatism (TIA) versus surgically induced astigmatism (SIA). (**I**) The angle of error of refractive astigmatism at 12 months postoperatively.

**Table 1 jcm-10-03067-t001:** Demographics and risk factors of patients with diffuse lamellar keratitis following LASIK.

	DLK (*n* = 637)		No DLK (*n* = 14348)		
	**Mean ± SD**	**Range**	**Mean ± SD**	**Range**	***p* Value**
Age, year	33.7 ± 8.25	(18, 57)	34.4 ± 8.81	(18, 85)	0.114
	***n***	**Percent**	***n***	**Percent**	
Gender [M/F]	243/193	(55.7%, 44.3%)	3838/3593	(51.6%, 48.4%)	0.107
Affected Eye [OD/OS]	329/308	(51.6%, 48.4%)	7219/7129	(50.3%, 49.7%)	0.536
	**Mean ± SD**	**Range**	**Mean ± SD**	**Range**	
Spherical Equivalent	−3.33 ± 2.27	(−9.50, 4.25)	−3.48 ± 2.14	(−11.25, 4.38)	0.081
Myopia ^a^	−3.53 ± 2.06	(−9.50, −0.50)	−3.64 ± 1.99	(−11.25, −0.50)	0.182
Hyperopia ^b^	1.97 ± 1.15	(0.50, 4.25)	1.63 ± 0.92	(0.50, 4.38)	0.115

OD—right eye; OS—left eye; M—male; F—female; y—years; ^a^—myopia defined by spherical equivalent ≤ −0.50 D; ^b^—hyperopia defined by spherical equivalent ≥ +0.50.

**Table 2 jcm-10-03067-t002:** DLK incidence after LASIK by femtosecond laser.

	Total Eyes	No DLK	Stage I	Stage II	Stage III	Stage IV
	*n*	*n* (%)	*n* (%)	*n* (%)	*n* (%)	*n* (%)
AMO iFs	12,512	11,984 (95.8)	435 (3.5)	66 (0.5)	24 (0.2)	3 (0.02)
WL FS200	906	841 (92.8)	59 (6.5)	6 (0.7)	0 (0.0)	0 (0.0)
Zeiss Visumax	1567	1523 (97.2)	39 (2.5)	4 (0.3)	1 (0.1)	0 (0.0)
Total	14,985	14,348 (95.7)	533 (3.6)	76 (0.5)	25 (0.2)	3 (0.02)

**Table 3 jcm-10-03067-t003:** Demographics comparison between different grades of DLK.

	Stage I (*n* = 533)	Stage II (*n* = 76)	Stage III (*n* = 25)	Stage IV (*n* = 3)	
Variable	Mean ± SD	Mean ± SD	Mean ± SD	Mean ± SD	*p* Value
Age, y	33.31 ± 8.09	33.79 ± 7.37	32.72 ± 8.85	28.33 ± 2.89	0.710
Time to onset d	1.34 ± 1.86	1.22 ± 0.81	1.48 ± 1.12	3 ± 1.73	0.579
Time to resolution, d	7.7 ± 7.59	9.62 ± 5.62	8.72 ± 5.21	10.67 ± 2.31	0.063
	***n***	***n***	***n***	***n***	
Gender [M/F]	298/235	43/33	12/13	2/1	0.854
Affected Eye [OD/OS]	277/256	39/37	11/14	1/2	0.830
	***n***	***n***	***n***	***n***	
AMO iFs	435	66	24	3	0.637
WL FS200	59	6	0	0	1
Zeiss VisuMax	39	4	1	0	1

DLK—diffuse lamellar keratitis; OD—right eye; OS—left eye; M—male; F—female; y—years; d—days.

**Table 4 jcm-10-03067-t004:** Treatments implemented between DLK stages.

	Stage I	Stage II	Stage III	Stage IV
	*n* (%)	*n* (%)	*n* (%)	*n* (%)
**Steroids**				
Prednisolone, topical *	530 (99.4)	75 (98.7)	22 (88.0)	3 (100)
Difluprednate (Durezol *), topical ***	80 (15.0)	50 (65.8)	17 (33.3)	1 (33.3)
Dexamethasone, topical *	0 (0)	2 (2.6)	2 (8.0)	0 (0)
Methylprednisolone, oral ***	8 (1.5)	11 (14.5)	13 (52.0)	2 (66.7)
**Antibiotic-Steroid Combinations**				
Tobramycin/Dexamethasone, topical	4 (0.8)	0 (0)	1 (4.0)	0 (0)
Tobramycin/Loteprednol etabonate (Zylet *), topical	1 (0.2)	0 (0)	0 (0)	0 (0)
**Antibiotics**				
Ofloxacin, topical *	419 (78.6)	64 (84.2)	17 (68)	0 (0)
Moxifloxacin, topical	137 (25.7)	17 (22.4)	10 (40.0)	1 (33.3)
Polytrim, topical *	0 (0)	2 (2.6)	0 (0)	0 (0)
Azithromycin, topical	2 (0.4)	0 (0)	0 (0)	0 (0)
Doxycylcine, oral ***	23 (4.3)	9 (11.8)	2 (8.0)	2 (66.7)
**Supplements**				
Vitamin C ***	0 (0)	4 (5.3)	4 (16.0)	2 (66.7)
Coenzyme Q10 ***	0 (0)	1 (1.3)	4 (16.0)	0 (0)
Multivitamin*	0 (0)	0 (0)	1 (4.0)	0 (0)
**Ocular Surface Treatments and Protection**				
Bandage Contact Lens ***	5 (0.9)	7 (9.2)	4 (16.0)	0 (0)
Shield *	0 (0)	2 (2.6)	0 (0)	0 (0)
Gel tears ***	15 (2.8)	10 (13.2)	4 (16.0)	0 (0)
Muro-128 ointment	0 (0)	1 (1.3)	0 (0)	0 (0)
Punctal Plugs	2 (0.4)	0 (0)	0 (0)	0 (0)
**Irrigation** *******	**8 (1.5)**	**11 (14.5)**	**20 (80.0)**	**1 (33.3)**

DLK—diffuse lamellar keratitis; *—*p* value ≤ 0.05; ***—*p* value ≤ 0.001. Bolded items indicate the general type of treatment used with specific treatments listed below.

**Table 5 jcm-10-03067-t005:** Studies comparing DLK following femtosecond laser flap creation.

	Year	FS Laser Model	FS Laser Frequency (kHz)	Total Eyes	Raster Energy (µJ)	Side Cut Energy (µJ)	Flap Thickness (µm)	Spot Separation (µm)	Postop Steroids	DLK Incidence *n* (%)
Binder [5]	2004	IntraLase FS15	15	103	1.9–2.8	6.0–8.0	110–140	10.0–14.0	-	20 (19.5%)
Javaloy [6]	2007	IntraLase FS15	15	100	1.6	2.3	120	-	QID	17 (17%)
Gil-Cazorla [7]	2008	IntraLase FS15	15	1000	1.7	1.9	120	-	8x/day	5 (0.5%)
Choe [8]	2010	IntraLase FS15 IntraLase FS30 Intralase FS60	15 30 60	176 180 164	1.9–3.3 1.9–2.3 1.6–1.9	2.9–4.2 2.1–3.2 2.0–2.3	100–130 100–130 100–130	9.0–11.0 9.0 8.0–9.0	QID	17 (9.7%) 24 (13.3%) 23 (14.0%)
Moshirfar [9]	2010	IntraLase FS60	60	902	1.15	0.8	110	-	QID	96 (10.6%)
De Paula [10]	2012	IntraLase FS60	60	801	1.4–1.8	1.6–2.4	102–131	-	-	99 (12.4%)
Tomita [11]	2013	IntraLase FS60 Ziemer Femto LDV	60 1000	304 514	1.0 0.1	0.8 0.8	- -	>1 <1	-	114 (37.5%) 42 (8.2%)
Kohnen [12]	2016	IntraLase FS60	60	1210	0.8	0.75	100-120	8	5x/day	89 (7.4%)
Torky [13]	2017	VisuMax	500	30	-	-	100	-	-	2 (6.6%)
Leccisotti [14]	2021	Ziemer LDV Z2/Z4	>5000	37,315	“nJ level”	-	95-110	-	QID	236 (0.6%)
Moshirfar	2021	AMO iFs WaveLight FS200 VisuMax	150 200 500	13,019 1031 1683	0.85–0.95 0.8 0.6	0.6 0.8 0.6	100–110 110–115 100–110	6.0 8.0 4.0	6x/day	528 (4.2%) 65 (7.2%) 44 (2.8%)

DLK—diffuse lamellar keratitis; FS—femtosecond; QID—four times per day.

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
