# Peer review of "Five-Year Incidence, Management, and Visual Outcomes of Diffuse Lamellar Keratitis after Femtosecond-Assisted LASIK"

_jcm, 2021, doi:10.3390/jcm10143067_

Round 1

Reviewer 1 Report

The authors of the underlying paper 

"Five-year Incidence, Management, and Visual Outcomes of Diffuse Lamellar Keratitis after Femtosecond-Assisted LASIK" present interesting data based on a very large number of laser vision corrections performed.  However, the manuscript could be strengthen addressing and revising the following issues:    

  • please check statistics, e.g. Tab.1 displays gender distribution of 58,4 % and 51,6 % that add up to 110 %
  • Analyzing all data should separate hyperopes and myopes, e.g. mean spherical equivalent counting myopes and hyperopes together does not say anything about the amount of ablation etc.  
  • Revise and check data in Tab3. Stage I n= 533 does not match to 427+59+39  fs laser procedures, also Stage II n=76 does not match to 70+5+4 fs laser procedures and Stage III n= 25 does not match to 25+0+1 fs laser procedures
  • Fig. 3 lacks a control group, in addition, eyes with DLK should be stratified in the four groups according to stage (I - IV) and these results should be compared with each other

Reviewer 2 Report

Dear Editor,

Thank you to provide me the opportunity to contribute to your journal. The authors submitted a study about a very interesting topic such as the study of diffuse lamellar keratitis after femto LASIK.

The authors provided a very interesting 5 years study comparing different femtosecond lasers with very good methodology. Some changes are required before accepting this manuscript.

Minor issues

Lines 49-51, please rephrase these sentences because it is not so clear.

Lines 54-60, authors could avoid historical information and concentrate more about the advantages of currently available FS machines.

Major issues

Figure 3 is too confusing, I suggest providing only 12 months data in order to let the readers easier understand the meaning of the study.

Please move table 5 reference on the discussion, it is not properly located in the results section.

Figure 4 can be avoided.

Round 2

Reviewer 1 Report

ok